# The 2023 global warming spike was driven by El Niño/Southern Oscillation

Shiv Priyam Raghuraman[1,2], Brian Soden[1], Amy Clement[1], Gabriel Vecchi[3,4], Sofia Menemenlis[5], Wenchang Yang[3]

[1]Dept. of Atmospheric Sciences, Rosenstiel School of Marine, Atmospheric, and Earth Science, University of Miami, Miami, 33149, USA
[2]Department of Climate, Meteorology & Atmospheric Sciences, University of Illinois Urbana-Champaign, Urbana, 61801, IL, USA
[3]Department of Geosciences, Princeton University, Princeton, 08544, NJ, USA
[4]High Meadows Environmental Institute, Princeton University, Princeton, 08544, NJ, USA
[5]Program in Atmospheric and Oceanic Sciences, Princeton University, Princeton, 08540, USA

*Correspondence to*: Shiv Priyam Raghuraman (sraghur2@illinois.edu)

**Abstract.** Global-mean surface temperature rapidly increased $0.29 \pm 0.04$ K from 2022 to 2023. Such a large interannual global warming spike is not unprecedented in the observational record with a previous instance occurring in 1976-77. However, why such large global warming spikes occur is unknown and the rapid global warming of 2023 has led to concerns that it could have been externally driven. Here we show that climate models that are subject only to internal variability can generate such spikes, but they are an uncommon occurrence ($p = 1.6 \pm 0.1\%$). However, when a prolonged La Niña immediately precedes an El Niño in the simulations, as occurred in nature in 1976-77 and 2022-23, such spikes become much more common ($p = 10.3 \pm 0.4\%$). Furthermore, we find that nearly all simulated spikes ($p = 88.5 \pm 0.3\%$) are associated with El Niño occurring that year. Thus, our results underscore the importance of El Niño/Southern Oscillation in driving the occurrence of global warming spikes such as the one in 2023, without needing to invoke anthropogenic forcing, such as changes in atmospheric concentrations of greenhouse gases or aerosols, as an explanation.

## 1 Introduction

Global-mean surface temperatures (GMST) have been rising since 1850 and more rapidly since the mid-20th century, principally because of human activities (IPCC, 2021). Observational (Lenssen et al., 2019; Morice et al., 2021; Rohde and Hausfather, 2020) analyses showed that GMST reached its highest recorded value in 2023, making it the warmest year on record. The rapid increase in annual-mean GMST of $0.29 \pm 0.04$ K (average of three observational datasets; Appendix A) in 2023 relative to 2022, an increase that occurs over one-two decades usually, has not only been a cause for concern societally but also scientifically as its causes were not obvious (Esper et al., 2024; Jiang et al., 2024; Kuhlbrodt et al., 2024; Rantanen and Laaksonen, 2024; Schmidt et al., 2024). Potential causes for this year-on-year spike include anthropogenic reasons such as greenhouse gas increases and aerosol pollution reductions, or natural reasons such as increased solar activity, volcanic-

induced stratospheric water vapor increases, and natural climate variability such as the El Niño/Southern Oscillation phenomenon (ENSO) (Schmidt et al., 2024). Most studies have focused on the external forcing aspects, particularly the role of aerosol pollution reductions, rather than quantifying the role of internal variability (Gettleman et al., 2024; Quaglia and Visioni, 2024; Schoeberl et al., 2024; Watson-Parris et al., 2024; Yoshioka et al., 2024; Zhang et al., 2024). This study focuses on the latter, and we will argue that ENSO is the primary reason for global warming spikes.

ENSO is a mode of internal variability in the climate system that comprises of a positive phase, El Niño, and a negative phase, La Niña (Trenberth, 1997). El Niño or La Niña occurs every few (three to seven, typically) years in the tropical Pacific Ocean and encompasses a global-scale rearrangement of temperatures, winds, sea level pressures, atmospheric convection, clouds, moisture, and radiation (Trenberth, 1997; Clement et al., 1996; Peng et al., 2024; Raghuraman et al., 2019; Soden, 1997). El Niño brings anomalous warmth to the Central and Eastern Pacific Ocean, and to other parts of the tropics with a lag, which increases GMST, and vice-versa for La Niña (Mann and Park, 1994; Mann et al., 2000). However, the degree of association of ENSO with global warming spikes has not yet been shown. An El Niño event occurred in 2023, which was preceded by a prolonged period of La Niña conditions from 2020-2022.

In the observational record since 1950, 2023 is not the only year with a global warming spike of this magnitude (an increase in interannual GMST greater than 0.25K (Appendix A)) to have occurred, 1977 too had a spike ($0.31 \pm 0.04\ K$). Both of these spikes occurred during an El Niño year and after a prolonged La Niña (1973-1976 and 2020-2022) (Fig. 1a). The spatial distribution of the 2023 spike resembles the canonical El Niño spatial pattern (Fig. 1b) (Peng et al., 2024). Thus, 2023 isn't unprecedented in producing a spike, and the observational record suggests a strong correlation between global warming spikes and ENSO (of the four long La Niña-El Niño transitions since 1950, two have led to spikes, i.e., p=50%). However, given the short record (74 years) it is difficult to draw conclusions based on a *post hoc* analysis of just two events. As a result, we turn to all available multi-centennial to multi-millennial global climate model simulations spanning 58,021 years across 64 models with no human influence ("piControl"; Table A1) (Eyring et al., 2016; Delworth et al., 2006; Gnanadesikan et al., 2006; Vecchi et al., 2014; Rugenstein et al., 2019).

In each model, we quantify the probability of a spike ($p(spike)$; Eq. (A1)), the probability of a spike occurring given a long La Niña to El Niño transition ($p(spike|Long\ La\ Niña + El\ Nino)$; Eq. (A2)), the probability of a spike occurring given a long La Niña occurring in prior years ($p(spike|Long\ La\ Niña)$; Eq. (A3)), the probability of a spike occurring given an El Niño occurring that year ($p(spike|El\ Niño)$; Eq. (A4)), and the probability of a spike associated with an El Niño occurring during the year ($p(El\ Nino|spike)$; Eq. (A5)). In the following sections we quantify the critical role ENSO plays in generating global warming spikes (Sec. 2) and present our conclusions (Sec. 3). Throughout our study we focus on the spike/interannual GMST change, rather than the record that a particular year may set.

## 2 Results

We find that spikes happen $1.6\% \pm 0.1\%$ (Multi-model means (MMM)) of the time on average in unforced model simulations ($p(spike)$ in Fig. 1c). The models show little inter-model spread with a minimum-maximum range of $p(spike)$ of 0-9%. That is, spikes are uncommon but can occur solely from internally generated climate variability. Given a long La Niña in the years prior to the spike followed by an El Niño during the spike year, the probability of a spike increases over six-fold (compared with unconditional probability $p(spike)$) to $10.3\% \pm 0.4\%$ on average in models (MMM; Fig. 1c's $p(spike|Long\ La\ Niña + El\ Niño)$). Thus, global warming spikes become much more likely during El Niño events preceded by a long La Niña – even if they are not to be expected ($p$=10.3%) and internal variability can produce such large spikes in GMST without invoking external forcing. The models show considerable inter-model spread with a minimum-maximum range of 0-52%, i.e., one model suggests no impact of a long La Niña to El Niño transition generating a spike while another suggests a one-in-two chance of a spike occurring given a prolonged La Niña to El Niño transition.

In addition to the impact a long La Niña to El Niño transition has on spikes, the individual impact of a long La Niña or an El Niño on a spike is quantified below. Given a long La Niña in the years prior to the spike, the probability of a spike amounts to $6.5\% \pm 0.3\%$ on average in models (MMM; Fig. 1c's $p(spike|Long\ La\ Niña)$). Similarly, given an El Niño during the spike year, the probability amounts to $6.3\% \pm 0.2\%$ on average in models (MMM; Fig. 1c's $p(spike|El\ Niño)$). The models show less intermodel spread in $p(spike|El\ Niño)$ compared to $p(spike|Long\ La\ Niña)$. Overall, the probability that a long La Niña or an El Niño can help generate a spike individually is lower than when the two are combined as a sequence of events. This shows the importance of how a long La Niña transition to an El Niño can increase the odds of a global warming spike.

So, ENSO can substantially increase the odds of warming spikes, but is ENSO a dominant driver of spikes? To explore this question, we compute the probability that El Niño events co-occur with a spike ($p(El\ Niño|spike)$). Spikes show a strong association with an El Niño occurring that year: the percentage of spikes associated with El Niño conditions is $88.5\% \pm 0.3\%$ on average in models (MMM; Fig. 1c's $p(El\ Niño|spike)$). Thus, virtually all spikes are associated with El Niño conditions that year. In fact, in over half of the models (38/64), the spike is always associated with El Niño conditions during the year, i.e., this probability is 100%. One example of this is the NOAA GFDL CM4 model where each of its spikes are associated with an El Niño event occurring during the year of the spike. This El Niño signal is clearly seen in the spatial pattern of one of the spikes in Fig. 1d. This fully coupled climate model has freely evolving sea surface temperatures, i.e., independent from 2023 observations, and yet its internally-generated spike's spatial pattern shows striking resemblance to the observed 2023 spike's spatial pattern (Fig. 1b,d): warming in the Central-East Pacific, cooling-warming dipole in the South Pacific, and warming in the Atlantic, Arctic, Africa, and Australia.

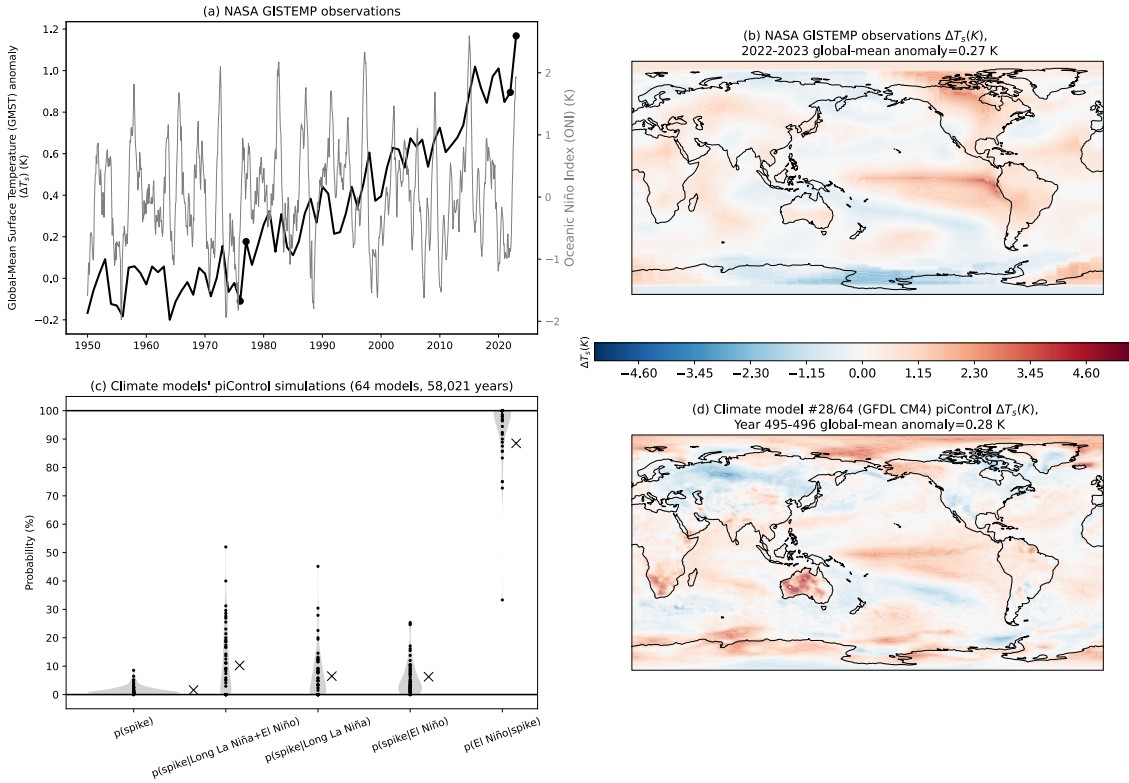

**Figure 1:** a. Annual-mean global-mean surface temperature (GMST) anomalies (baseline 1951-1980; black) and monthly-mean Oceanic Niño Index (detrended; grey) from NASA GISTEMP observations. Dots represent GMST spikes ($\Delta$GMST>0.25 K) from 1976 to 1977 and 2022 to 2023. b. Spatial pattern of surface temperature change from 2022 to 2023, i.e., 2023 spike, from NASA GISTEMP observations. c. Probabilities based on Eq. (A1) - (A5). Dots denote each model and crosses denote the multi-model mean (MMM). d. Spatial pattern of a surface temperature change from Year 495 to Year 496 in one of the 64 models' piControl simulations analyzed (GFDL CM4) is provided as an example.

## 2.1 Sensitivity tests

Below, we test the sensitivity of our results to choices in the ENSO metric, the annual-mean definition, and the observational dataset (Table 1). We find that our results remain robust. First, we use an alternative ENSO metric, the relative Niño3.4 index, to test for the impact of different ENSO amplitudes/definitions (Van Oldenborgh et al., 2021). The MMM $p(spike|Long\ La\ Ni\tilde{n}a + El\ Ni\tilde{n}o)$ is 10.3% $\pm$ 0.4% for the regular Nino3.4 metric and 10.3% $\pm$ 0.8% for the relative Nino3.4 index, i.e., identical values on average. Note that $p(spike)$ remains unchanged by definition, as it is an unconditional probability, i.e., independent of ENSO.

Second, due to a lag between ENSO and GMST, we computed the probabilities using a September-August annual-mean definition to test if the influence of ENSO on spikes changes. The MMM $p(spike) = 2.3\%$ and $p(spike|Long\ La\ Niña + El\ Niño) = 17.4\%$, compared with the regular January-December annual-mean definition probabilities of 1.6% and 10.3%, respectively. This implies that the probability increases over seven-fold, compared to over six-fold in the regular definition. Thus, the September-August annual-mean definition has a larger influence on spikes as El Niño continues to impact GMST even the following year.

Third, we test how sensitive our results are to the choice of the individual observational dataset and its uncertainty. GISTEMP has a slightly smaller spike and a slightly larger uncertainty when compared with the average of the three datasets, resulting in a smaller spike threshold. This yields larger probabilities: MMM $p(spike) = 2.9\%$ and $p(spike|Long\ La\ Niña + El\ Niño) = 16.8\%$. HadCRUT5 and Berkeley Earth Surface Temperature have slightly larger spikes and equal or slightly smaller uncertainties when compared with the average of the three datasets, resulting in a larger spike threshold. This yields smaller probabilities: MMM $p(spike) = 1.3\%$ and $1.0\%$ , respectively, and $p(spike|Long\ La\ Niña + El\ Niño) = 8.3\%$ and $6.8\%$, respectively. The change in $p(spike)$ can be visualized in the probability distribution in Figure A1: an increase in the spike threshold value (going further right on the x-axis) reduces the probability of a spike as the magnitude of the spike increases due the Gaussian nature of the distribution. Overall, the average of these 3 probabilities is 10.6%, nearly identical to the probability computed based on the average of the 3 spike definitions (10.3% $\pm$ 0.4 %), placing confidence in our methods. Furthermore, in all three datasets, the six-fold increase in the probability is maintained.

**Table 1**: Sensitivity of results to choices in the ENSO metric (relative Niño3.4, Van Oldenborgh et al., 2021), the annual-mean definition (September-August), and the observational dataset (GISTEMP, HadCRUT5, and Berkeley Earth Surface Temperature). Multiplicative factor refers to the ratio $\frac{p(spike|Long\ La\ Niña + El\ Niño)}{p(spike)}$.

| Sensitivity parameter | Spike ( $\Delta T_s$ ; Interannual GMST change) (K) | Spike Threshold (K) | $p(spike)$ (%) | $p(spike|Long\ La\ Niña + El\ Niño)$ (%) | Multiplicative factor (unitless) |
|---|---|---|---|---|---|
| Relative Niño3.4 | $0.29 \pm 0.04$ | 0.25 | 1.6 | 10.3 | 6.4 |
| Sep-Aug annual-mean | $0.29 \pm 0.04$ | 0.25 | 2.3 | 17.4 | 7.6 |
| GISTEMP | $0.27 \pm 0.05$ | 0.22 | 2.9 | 16.8 | 5.8 |
| HadCRUT5 | $0.30 \pm 0.04$ | 0.26 | 1.3 | 8.3 | 6.4 |
| Berkeley | $0.30 \pm 0.03$ | 0.27 | 1.0 | 6.8 | 6.8 |

**3 Conclusions and Discussion**

Our results show that global warming spikes can happen without any human influence. Such global warming spike events seem uncommon when unconditioned on ENSO history. But when conditioned on a long La Niña to El Niño transition occurring, these global warming spikes become much more common. We underscore that our findings regarding the association of global warming spikes with ENSO does not undermine the vast body of literature on how anthropogenic activities are causing long-term global warming (IPCC, 2021). However, ENSO variability against a background warming trend may lead to year-on-year spikes that are also historical temperature records (Forster et al., 2024; Min, 2024).

Previous work concluded that it's extremely unlikely that internal variability alone can explain the September 2023 GMST spike (Rantanen and Laaksonen, 2024; hereafter RL24). However, our results put 2023 temperatures into broader context, and emphasize that internal variability plays a central role in explaining the annual-mean temperature spike. The apparent contrast between our conclusions and those of RL24 arise from differences in our approaches to the analysis. RL24 focus on a single month and define a spike/jump as relative to the previous record (September 2020). Temperatures across the multi-year gaps between monthly records may be influenced by different factors such as lower frequency variability or anthropogenic forcing. By contrast, we focus on the annual-mean and define a spike as relative to the previous year, considering continuous transitions that can be related to interannual variability. They use forced simulations, while we use unforced simulations and an order of magnitude of more data. They consider only the unconditional probability, for which the probability of a spike is divorced from the underlying atmosphere-ocean-climate processes. We compute the conditional probability, which reveals the central role of ENSO in explaining year-to-year temperature spikes. Regarding the September 2023 spike, RL24 find that the September 2023 GMST beat its previous record by 0.5 K and this margin is outside the realm of internal variability (~1% probability). We find a similar result with our methodology of GISTEMP's GMST in September 2023 increasing 0.59 K relative to September 2022 and piControl simulations showing this spike being exceptionally unlikely: $p(spike_{Sep}) = 0.01\%$. However, we also find other such examples of small probabilities (<1% probability) in other months and years outside of 2023: models simulate spikes of the magnitude of February 1994-1995's with a probability $p(spike_{Feb}) = 0.13\%$ and May 1976-1977's with a probability of $p(spike_{May}) = 0.1\%$.

Looking forward to 2024, our unforced climate models simulations can provide some perspective on how likely another spike in GMST will be. We find that the probability there are two back-to-back spikes in the models is 0.02%. Thus, back-to-back spikes are rare, but when they do occur, we find that it is often associated with a long El Niño. Tropical Pacific

conditions have turned neutral over 2024 ((https://www.climate.gov/news-features/blogs/enso/september-2024-enso-update-binge-watch), suggesting that the probability of another global warming spike (another 0.25 K increase or more in GMST) in 2024 is low. Looking further forward, model projections diverge on whether there will be an increase or decrease in the number of El Niños and long La Niñas due to greenhouse gas warming (Cai et al., 2015; DiNezio et al., 2012; Vecchi et al., 2008). If the probability of spikes given these ENSO events remains the same, this would imply that in the future, the number of global warming spikes increases or decreases depending on ENSO frequency changes (Eq. (A6)). Finally, future research should quantify the impact of other forms of internal variability such as the Atlantic Multidecadal Oscillation (Li et al., 2024), and its relation/co-occurrence with ENSO (Fig. 1b,d show similar warming patterns in the Atlantic), on the 2023 spike.

## Appendix A: Methods

We define a spike as a year-to-year change in GMST ($\Delta T_s$; Fig. A1) that exceeds 0.25 K. This value is based on the 2023 increase in GMST relative to 2022 being $0.29 \pm 0.04$ K (average of GISTEMP, HadCRUT5, BEST estimates (Lenssen et al., 2019; Morice et al., 2021; Rohde and Hausfather, 2020); 95% anomaly uncertainty). Thus, 0.25 K is a lower bound. The piControl simulations in models are fully coupled simulations that have freely evolving temperatures with no human influence. We use models' full time series and only those that span at least 500 years. Climate models differ in their representations of ENSO, and this may impact the probabilities we compute for each model. This is why we analyze all available climate models (64), not just a subset. Furthermore, we analyzed models not only in this generation (CMIP6) but also some models from previous generations (CMIP3 and CMIP5). Multi-model means (MMM) are reported by weighting by each model's time series length. Simple averaging yields similar results. Uncertainties are reported as 95% confidence intervals, i.e., $1.96 \times \frac{\sigma}{\sqrt{n}}$ where $\sigma$ is the standard deviation of a probability across models and $n$ is the number of models.

We define a long La Niña event to be when the detrended Oceanic Niño Index (ONI) exceeds $-0.5$ K for at least 18 consecutive months (this threshold was chosen to mimic the conditions leading up to 2023). The ONI is defined as the three-month running mean of sea surface temperature monthly anomalies in the Niño3.4 region, a Central Pacific region spanning 5°S-5°N, 190°E-240°E, and is widely used for defining ENSO events (https://origin.cpc.ncep.noaa.gov/products/analysis_monitoring/ensostuff/ONI_v5.php). We define an El Niño event as when the detrended ONI exceeds 0.5 K for at least 5 consecutive months. A long La Niña to El Niño transition is defined as one that occurs in less than a year.

The probability of a spike is given by:

$$p(spike) = \frac{Number\ of\ spikes}{Number\ of\ years\ in\ time\ series} \tag{A1}$$

The probability of a spike given a sequence of a long La Niña event occurring in prior years followed by an El Niño event occurring the year of the spike can be expressed as a conditional probability:

$$p(spike|Long\ La\ Niña + El\ Niño) = \frac{p(spike \cap Long\ La\ Niña + El\ Niño)}{p(Long\ La\ Niña + El\ Niño)} \tag{A2a}$$

$$p(spike|Long\ La\ Niña + El\ Niño) = \frac{Number\ of\ spikes\ that\ follow\ Long\ La\ Niña + El\ Niño\ transitions}{Number\ of\ Long\ La\ Niña + El\ Niño\ transitions} \tag{A2b}$$

Similarly, the probability of a spike given a long La Niña event occurring in prior years (the end of the event must be less than a year from the spike year) can be expressed as a conditional probability:

$$p(spike|Long\ La\ Niña) = \frac{p(spike \cap Long\ La\ Niña)}{p(Long\ La\ Niña)} \tag{A3a}$$

$$p(spike|Long\ La\ Niña) = \frac{Number\ of\ spikes\ that\ follow\ a\ Long\ La\ Niña}{Number\ of\ Long\ La\ Niñas} \tag{A3b}$$

Similarly, the probability of a spike given an El Niño event occurring that year can also be expressed as a conditional probability:

$$p(spike|El\ Niño) = \frac{p(spike \cap El\ Niño)}{p(El\ Niño)} \tag{A4a}$$

$$p(spike|El\ Niño) = \frac{Number\ of\ spikes\ during\ El\ Niño\ year}{Number\ of\ El\ Niños} \tag{A4b}$$

The probability of a spike being associated with El Niño conditions, i.e., the percentage of spikes associated with El Niño conditions, can also be expressed as a conditional probability:

$$p(El\ Niño|spike) = \frac{p(El\ Niño \cap spike)}{p(spike)} \tag{A5a}$$

$$p(El\ Niño|spike) = \frac{Number\ of\ spikes\ during\ an\ El\ Niño\ year}{Number\ of\ spikes} \tag{A5b}$$

We plot Equations (A1)-(A5)'s values for each climate model in Fig. 1c. Note that Equations (A4) and (A5) can be related via Bayes' Theorem:

$$p(spike|El\ Niño) = \frac{p(El\ Niño|spike) \times p(spike)}{p(El\ Niño)} \tag{A6}$$

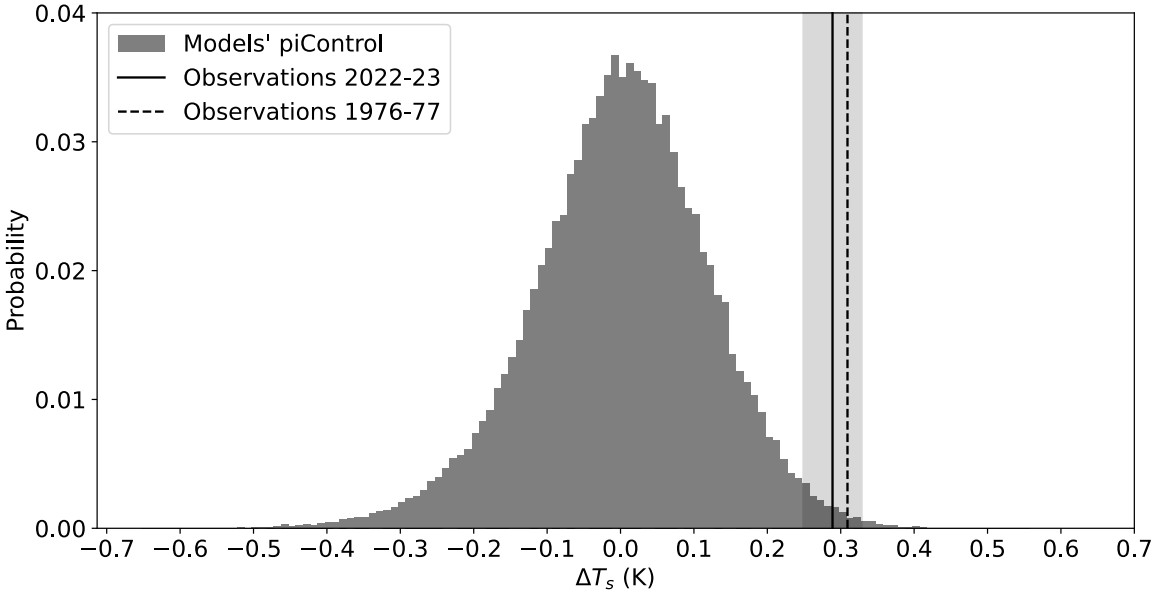

220

**Figure A1**: Probability distribution of year-to-year change in GMST ($\Delta T_s$) in all piControl simulations in 64 models spanning 58,021 years. Mean and standard deviation are 0 and 0.12 K, respectively. The shaded area represents the $\pm 0.04$ K uncertainty in the 2022-2023 observed annual-mean GMST anomaly of 0.29 K. Simulated $\Delta T_s$ within and to the right of this shaded region represent the probability of global warming spikes ($p(spike)$). This is an unconditional probability, i.e., independent of ENSO.

225

**Table A1**: piControl models and number of years for monthly-mean surface temperature ('ts'). Only for GFDL CM2.1, FLOR, and CCSM3 do we exclude the first 20 years due to particularly spurious model drift. Centennial-millennial length drifts are inconsequential for $\Delta T_s$ as spikes are defined as interannual changes and are accounted in the ONI by detrending.

|  | Model name | Realization | Number of years |
|---|---|---|---|
|  | CMIP6 piControl |  |  |
| 1. | ACCESS-CM2 | r1i1p1f1 | 500 |
| 2. | ACCESS-ESM1-5 | r1i1p1f1 | 1000 |
| 3. | AWI-CM-1-1-MR | r1i1p1f1 | 500 |
| 4. | BCC-CSM2-MR | r1i1p1f1 | 600 |
| 5. | CAMS-CSM1-0 | r1i1p1f1 | 500 |
| 6. | CanESM5 | r1i1p1f1 | 1000 |
| 7. | CanESM5-1 | r1i1p1f1 | 500 |
| 8. | CanESM5-CanOE | r1i1p2f1 | 501 |
| 9. | CAS-ESM2-0 | r1i1p1f1 | 550 |
| 10. | CESM2 | r1i1p1f1 | 1200 |
| 11. | CESM2-FV2 | r1i1p1f1 | 500 |

| 12. | CESM2-WACCM | r1i1p1f1 | 499 |
|---|---|---|---|
| 13. | CESM2-WACCM-FV2 | r1i1p1f1 | 500 |
| 14. | CIESM | r1i1p1f1 | 500 |
| 15. | CMCC-CM2-SR5 | r1i1p1f1 | 500 |
| 16. | CMCC-ESM2 | r1i1p1f1 | 500 |
| 17. | CNRM-ESM2-1 | r1i1p1f2 | 500 |
| 18. | E3SM-1-0 | r1i1p1f1 | 500 |
| 19. | E3SM-2-0 | r1i1p1f1 | 500 |
| 20. | E3SM-2-0-NARRM | r1i1p1f1 | 500 |
| 21. | EC-Earth3 | r1i1p1f1 | 501 |
| 22. | EC-Earth3-CC | r1i1p1f1 | 505 |
| 23. | EC-Earth3-Veg | r1i1p1f1 | 500 |
| 24. | EC-Earth3-Veg-LR | r1i1p1f1 | 501 |
| 25. | FGOALS-f3-L | r1i1p1f1 | 561 |
| 26. | FGOALS-g3 | r1i1p1f1 | 700 |
| 27. | FIO-ESM-2-0 | r1i1p1f1 | 500 |
| 28. | GFDL-CM4 | r1i1p1f1 | 500 |
| 29. | GFDL-ESM4 | r1i1p1f1 | 500 |
| 30. | GISS-E2-1-G | r1i1p1f1 | 851 |
| 31. | GISS-E2-1-H | r1i1p1f1 | 801 |
| 32. | HadGEM3-GC31-LL | r1i1p1f1 | 2000 |
| 33. | HadGEM3-GC31-MM | r1i1p1f1 | 500 |
| 34. | ICON-ESM-LR | r1i1p1f1 | 500 |
| 35. | INM-CM4-8 | r1i1p1f1 | 531 |
| 36. | INM-CM5-0 | r1i1p1f1 | 1201 |
| 37. | IPSL-CM6A-LR | r1i1p1f1 | 2000 |
| 38. | IPSL-CM6A-MR1 | r1i1p1f1 | 500 |
| 39. | MCM-UA-1-0 | r1i1p1f1 | 500 |
| 40. | MIROC6 | r1i1p1f1 | 800 |
| 41. | MIROC-ES2L | r1i1p1f2 | 500 |
| 42. | MPI-ESM-1-2-HAM | r1i1p1f1 | 1000 |
| 43. | MPI-ESM1-2-HR | r1i1p1f1 | 500 |

| 44. | MPI-ESM1-2-LR | r1i1p1f1 | 1000 |
|-----|---------------|----------|------|
| 45. | MRI-ESM2-0 | r1i1p1f1 | 701 |
| 46. | NESM3 | r1i1p1f1 | 500 |
| 47. | NorCPM1 | r1i1p1f1 | 500 |
| 48. | NorESM2-LM | r1i1p1f1 | 500 |
| 49. | NorESM2-MM | r1i1p1f1 | 501 |
| 50. | SAM0-UNICON | r1i1p1f1 | 700 |
| 51. | TaiESM1 | r1i1p1f1 | 500 |
| 52. | UKESM1-0-LL | r1i1p1f2 | 1880 |
|  | LongRunMIP Control |  |  |
| 53. | CCSM3 | - | 1510 |
| 54. | CESM104 | - | 1000 |
| 55. | CNRM-CM6-1 | - | 2000 |
| 56. | EC-Earth | - | 508 |
| 57. | GFDL CM3 | - | 5200 |
| 58. | GFDL ESM2M | - | 1340 |
| 59. | HadCM3L | - | 1000 |
| 60. | IPSL-CM5A | - | 1000 |
| 61. | MIROC3.2 | - | 680 |
| 62. | MPI-ESM1.2 | - | 1237 |
|  | Other models' Control |  |  |
| 63. | GFDL CM2.1 | - | 3980 |
| 64. | GFDL FLOR | - | 2980 |

**Code availability**

Code can be accessed at Raghuraman (2024).

**Data availability**

The observed surface temperature data was obtained from https://data.giss.nasa.gov/gistemp/, https://www.metoffice.gov.uk/hadobs/hadcrut5/, and https://berkeleyearth.org/data/. CMIP6 piControl data was obtained from the CMIP6 archive (https://esgf-node.llnl.gov/projects/cmip6/). LongRunMIP data was obtained from

235 https://www.longrunmip.org/. CM2.1 and FLOR surface temperature data have been deposited in the Zenodo database (Raghuraman et al., 2024).

## Author contribution

SPR performed analysis and writing with regular feedback and inputs to the manuscript from all co-authors. GV, SM, WY performed the CM2.1 and FLOR simulations.

## Competing interests

The authors declare that they have no conflict of interest.

## Acknowledgements

We acknowledge the World Climate Research Programme, which, through its Working Group on Coupled Modelling, coordinated and promoted CMIP6. We thank the climate modeling groups for producing and making available their model
output, the Earth System Grid Federation (ESGF) for archiving the data and providing access, and the multiple funding agencies that support CMIP6 and ESGF. We thank the anonymous referee, Mika Rantanen, and Ales Kuchar for their comments.

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
