# Peer review of "The 2023 global warming spike was driven by El Niño/Southern Oscillation"

_EGUsphere, 2024_

## Author Comment (AC1)

RC1: 'Comment on egusphere-2024-1937', Anonymous Referee #1

This is a timely study exploring an important null hypothesis for explaining last year's extreme global warming. Although the approach is sound and the results solid I think that a bit more detail in the analysis will go a long way to rule out other explanations for the record-breaking GMTS in 2023. I really like the idea of quantifying spikes relative to a persistent La Nina - this idea was floated my Michael Mann on twitter so some credit should be give to him in the Acknowledgements.

We thank the referee for their review and encouragement. We have now added Mann and Park, 1994 and Mann et al., 2000 to the references and cited these papers in second paragraph of the paper.

Here are my suggestions for adding more depth or dimensions to the analysis:

In the end the El Nino of 2023 was not as strong as expected and therefore its effect on GMTS should not be compared with the events of 1997 and 2015 or with all events in models. I wonder if there is a sensitivity of the results to the magnitude of simulated ENSO. Perhaps figure 1A could be expanded to include PDFs for events of different magnitude. I suggest using other metrics of ENSO amplitude such as the SOI to avoid issues related to the warming trend on the definition of El Nino - see: https://iopscience.iop.org/article/10.1088/1748-9326/abe9ed

Thank you for this comment and suggestion. We use the referee's suggested reference (Van Oldenborgh et al., 2021) and use their metric of ENSO amplitude in our analysis. We find that our results are robust to using their relative Nino3.4 index: the MMM p(spike|Long La Niña+El Niño) is 10.3% $\pm$ 0.4% for the regular Nino3.4 metric and 10.3% $\pm$ 0.8% for the relative Nino3.4 metric of Van Oldenborgh et al. (2021) (note that these values are for a spike of 0.25 K or more, not 0.22 K as before (see Referee #2's comments and our responses below)). These findings are now added to the manuscript in a new section called "Sensitivity tests" (Section 2.1) and a new table (Table 1). Note that p(spike) remains unchanged by definition, as it is an unconditional probability, i.e., independent of ENSO.

Additionally, we note that we had been and continue to detrend the ONI (to avoid issues related to the warming trend on the definition of El Niño as rightly pointed out by the referee) for the regular Nino3.4 metric as mentioned in the Methods section and captions of Figure 1 and Table A1. Finally, we also wish to clarify that the probability distribution function (PDF) in Figure A1 does not use any ENSO definition; it is merely the distribution of annual-mean GMST interannual changes in climate models. We now make this clearer in the caption of Figure A1.

Also, the largest influence of El Nino on GMTS occurs on the year after the peak, not before. That should be 2024 not 2023. I wonder if a.more nuanced analysis should be performed to isolate the months when El Nino (and La Nina) have the largest influence on GMTS. I have performed this analysis using observations via lag-correlation of GMTS and Nino-34 and found that ENSO has the largest imnpact from Nov of year 0 to June of year +1. Perhaps GMTS should be computed for an "ENSO" year starting on Sep of year 0 and ending on August of year 1 for a more clear isolation of their correlation.

Thank you for this comment and suggestion. We computed the probabilities using your suggested Sep-Aug annual-mean definition and found that the probabilities indeed go up. For a spike of 0.25 K or more, p(spike)=2.3% and p(spike|Long La Niña+El Niño)=17.4% (compared with regular Jan-Dec annual-mean definition of 1.6% and 10.3%, respectively). This implies that the probability increases over seven-fold (compared to over six-fold in the regular case). Thus, the referee is right that this Sep-Aug annual-mean definition has a larger influence on GMST. These findings are now added to the manuscript in a new section called "Sensitivity tests" (Section 2.1) and a new table (Table 1).

Otherwise a very important study that I hope gets published soon.

We thank the referee again for their review and encouragement.

RC2: 'Comment on egusphere-2024-1937', Mika Rantanen

Review of Atmospheric Chemistry and Physics manuscript egusphere-2024-1937 "The 2023 global warming spike was driven by El Niño/Southern Oscillation" by Raghuraman et al.

This manuscript tackles a topical issue by attempting to explain the 2023 global warmth by El Nino/Southern Oscillation. The authors employ a large set of climate model ensembles and calculate the probabilities of global warming "spikes" during different phases of ENSO and also after long La Nina events. Although the spike in global temperature in 2023 was an uncommon occurrence, its probability is multiplied when an El Nino event follows a prolonged La Nina. I think the manuscript is well written and adds well to the ever-growing literature explaining the causes of the 2023 warmth. Therefore, I recommend the publication of the study after my minor comments have been addressed.

We thank the referee for their review and encouragement.

1.  I downloaded the NASA GISTEMP table of global mean temperatures from here: https://data.giss.nasa.gov/gistemp/, and for me the annual mean data (column J-D) is 1.17°C for 2023 and 0.89°C for 2022 (accessed 16 July 2024). That gives an increase of 0.28°C. In the paper, the authors say 0.27°C. Can you explain where you got this value? It's not a big deal, but since you say in the abstract that the increase is not unprecedented, even small differences are important. Also, 0.28°C would be the largest difference in the GISTEMP data, tied with 1977.

    Thank you for this comment. In the original gridded GISTEMP netcdf data that we used, 2022 = 0.895 K and 2023 = 1.167 K. Thus, the difference = 0.272 K and hence our use of 0.27 K for the GISTEMP spike. So, this appears to be a rounding issue on the webpage.

2.  That leads me to my 2nd comment, which is probably the most important. How sensitive are your results for the choice of observational dataset and its uncertainty? I quickly looked at how large was the year-to-year difference in other global temperature datasets from here: https://climate.metoffice.cloud/temperature.html. In 1950-2023, the year-to-year difference in 2023 is ranked 1-2 in HadCRUT5, Berkeley Earth, Gistemp, NOAA and ERA5 with values ranging between 0.28-0.30°C. For example, the difference between 2022 and 2023 in ERA5 and HadCRUT5 was 0.30°C. If the spike is defined by this number, the probability of it occurring would obviously be greatly reduced (see comment 8).

    Thank you for this comment and suggestion. We now consider the suggested additional observational datasets in the manuscript. The GMST annual-mean spikes in GISTEMP, HadCRUT5, and BEST are $0.27 \pm 0.05$ K, $0.30 \pm 0.04$ K, and $0.30 \pm 0.03$ K, respectively. NOAAGlobalTemp version 6 shows a spike of 0.29 K, but there is no estimate of uncertainty available yet (Dr. Xungang Yin of NOAAGlobalTemp, personal communication). In any case, this NOAA estimate is within the uncertainty of other estimates and in fact equals the mean of the other 3 estimates. This mean, $0.29 \pm 0.04$ K, is now used for the whole paper, including the abstract, as suggested by the referee. As a result, the numbers have changed throughout the text, but the findings remain robust. Indeed, as the referee notes, p(spike) decreases, to 1.6% (MMM), but the MMM p(spike|Long La Niña+El Niño) is $10.3\% \pm 0.4\%$ (probability multiplies over six-fold as before). We further list the probabilities for each of the 3 observational datasets' spike definitions in a new section called "Sensitivity tests" (Section 2.1) and a new table (Table 1). The average of these 3 probabilities is 10.6%, nearly identical to the probability computed based on the average of the 3 spike definitions (10.3% $\pm$ 0.4 %), placing confidence in our methods.

3.  Only the number of the 2023 spike was mentioned in the paper. I was also missing the numbers of the 1977 and 1957 spikes to put the 2023 value into context. They are in Fig. A1, but the numerical values seem not to be listed anywhere.

    Thank you for pointing this out. We now mention the magnitude of the 1977 spike (0.31 K) in Sec 1 paragraph 3. Note that we have removed 1957 from the manuscript as it is below the new spike threshold of 0.25 K.

4. L11. I think it's not unknown that ENSO modulates global temperature and causes "spikes" in global temperature, such as 2016 or 1998. Rather, I think it is a well known fact. I'd suggest rephrasing this part of the sentence.

   Thank you for this comment. We rephrased this to "such large global warming spikes". We wish to clarify that we did not state that it's unknown that ENSO modulates global temperature. In fact, we say this in the Introduction. We argue that it is unknown what causes "spikes" as defined by $0.29 \pm 0.04$ K. 2016 nor 1998 fit that definition.

5. L21-23. Relating to the comment 2, I find this a bit confusing. NASA GISTEMP is only one of the several observational datasets. The statement that GMST increased 0.27 +- 0.05 K is GISTEMP-specific, and other datasets might give slightly different numbers. I think you should not be so tied to GISTEMP in this context.

   Thank you for this comment. Please see our response to comment 2. Furthermore, we have added in the HadCRUT5 and BEST references and explicitly state "(average of three observational datasets)".

6. L96. I think Rantanen and Laaksonen (2024) stress in their paper that internal variability alone cannot explain the September 2023 record margin with high likelihood. This does not exclude the possibility that internal variability still explains most of the spike. To me, saying that internal variability has little power sounds like Rantanen and Laaksonen (2024) is attributing less than half of the spike to internal variability, when this is clearly not the case.

   Thank you for pointing this out. We have revised the text accordingly to "Previous work concluded that it's extremely unlikely that internal variability alone can explain the September 2023 GMST spike (Rantanen and Laaksonen, 2024; hereafter RL24).".

7. L110. I'd like to emphasize that your probabilities for Feb 1995 (0.13%) and May 1977 (0.1%) are still an order of magnitude higher than for September 2023 (0.01%). So calling them similar examples and 2023 being not unique sounds like a slight understatement. Also, El Nino tends to peak in NH winter, and thus I'd imagine that the global warming spikes tend to be stronger NH winter/spring, i.e. during or soon after the El Nino event. This was not the case in Sep 2023.

   Thank you for pointing this out. We have deleted the final sentence stating 2023 is not unique and removed "similar".

8. L126. Again, I find it a bit odd to define the spike using the lower bound of the GISTEMP uncertainty interval. Why not define the spike using the average of multiple observational datasets (HadCRUT5, GISTEMP, Berkeley, NOAA...)? I guess the main finding, i.e. the probability of the spike (p=2.6 %) is greatly dependent on how you define the spike. If the spikes were calculated as averages over several datasets, using the best estimate from each dataset (rather than the lower bound), the 2023 spike would probably be estimated to be around 0.29 K (this is the number that most people consider to be the difference between 2022 and 2023). In this case, the probability would be much lower than the 2.6% reported in your paper?

   Thank you for this comment. Please see our response to comment 2. Furthermore, no observational dataset is perfect due to numerous uncertainties such as spatial and temporal coverage uncertainties, measurement uncertainty, and parametric uncertainty. Thus, we must consider the uncertainty attached to each dataset rather than just the mean estimate. Therefore, the spike must consider the lower bound of the uncertainty. We quantify the probabilities for different spike thresholds according to each observational dataset in Table 1.

9. L138. "The ONI is defined as the sea surface temperature change". Change from which? Do you mean sea surface temperature anomaly from some certain baseline?

   Thank you for pointing this out. We have revised the text accordingly to "The ONI is defined as the three-month running mean of sea surface temperature monthly anomalies".

CC1: 'Comment on egusphere-2024-1937', Ales Kuchar

I recommend using violin plots in Fig. 1c in addition to dots for individual models. Violin plots are similar to box plots, except that they also show the probability density of the data at different values, usually smoothed by a kernel density estimator. While a box plot shows summary statistics such as mean/median and interquartile ranges, the violin plot shows the full distribution of the data. This would be useful to see how the individual models are distributed.

Thank you for your comment. We have now revised Fig. 1c to include violin plots in addition to dots for individual models.